# CO_2_ Utilization and Sequestration in Organic and Inorganic Nanopores During Depressurization and Huff-n-Puff Process

**DOI:** 10.3390/nano14211698

**Published:** 2024-10-24

**Authors:** Jiadong Guo, Shaoqi Kong, Kunjie Li, Guoan Ren, Tao Yang, Kui Dong, Yueliang Liu

**Affiliations:** 1College of Mining Engineering, Taiyuan University of Technology, Taiyuan 030024, China; 18536201839@163.com (J.G.); dongkui@tyut.edu.cn (K.D.); 2State Key Laboratory of Coaland CBM Co-Mining, Taiyuan 030032, China; marden2008@163.com; 3Jin Chuang Investment Co., Ltd., Taiyuan 030000, China; rga78919@163.com; 4School of Mine Safety, North China Institute of Science and Technology, Beijing 101601, China; yangtao585@163.com; 5State Key Laboratory of Petroleum Resources and Prospecting, China University of Petroleum (Beijing), Beijing 102249, China; 6College of Petroleum Engineering, China University of Petroleum (Beijing), Beijing 102249, China; 7College of Carbon Neutrality Future Technology, China University of Petroleum (Beijing), Beijing 102249, China

**Keywords:** shale gas, pressure drawdown, organic nanopores, CO_2_ huff and puff, CO_2_ sequestration

## Abstract

CO_2_ injection in shale reservoirs is more suitable than the conventional recovering methods due to its easier injectivity and higher sweep efficiency. In this work, Grand Canonical Monte Carlo (GCMC) simulation is employed to investigate the adsorption/desorption behavior of CH_4_-C_4_H_10_ and CH_4_-C_4_H_10_-CO_2_ mixtures in organic and inorganic nanopores during pressure drawdown and CO_2_ huff and puff processes. The huff and puff process involves injecting CO_2_ into the micro- and mesopores, where the system pressure is increased during the huffing process and decreased during the puffing process. The fundamental mechanism of shale gas recovery using the CO_2_ injection method is thereby revealed from the nanopore-scale perspective. During primary gas production, CH_4_ is more likely to be produced as the reservoir pressure drops. On the contrary, C_4_H_10_ tends to be trapped in these organic nanopores and is hard to extract, especially from micropores and inorganic pores. During the CO_2_ huffing period, the adsorbed CH_4_ and C_4_H_10_ are recovered efficiently from the inorganic mesopores. On the contrary, the adsorbed C_4_H_10_ is slightly extracted from the inorganic micropores during the CO_2_ puffing period. During the CO_2_ puff process, the adsorbed CH_4_ desorbs from the pore surface and is thus heavily recovered, while the adsorbed C_4_H_10_ cannot be readily produced. During CO_2_ huff and puff, the recovery efficiency of CH_4_ is higher in the organic pores than that in the inorganic pores. More importantly, the recovery efficiency of C_4_H_10_ reaches the highest levels in both the inorganic and organic pores during the CO_2_ huff and puff process, suggesting that the CO_2_ huff and puff method is more advanced for heavier hydrocarbon recovery compared to the pressure drawdown method. In addition to CO_2_ storage, CO_2_ sequestration in the adsorbed state is safer than that in the free state. In our work, it was found that the high content of organic matter, high pressure, and small pores are beneficial factors for CO_2_ sequestration transforming into adsorbed state storage.

## 1. Introduction

The potential for shale oil resources in China’s saline lake sediments is huge [1], which makes shale oil an important energy type, as the output of traditional fuels is declining sharply [2,3,4,5]. Although the original reserves of shale oil are abundant all over the world, commercially recoverable shale is quite limited due to its distinctive characteristics, such as extremely low permeability, etc. [6,7]. In recent years, horizontal drilling and multi-staged hydraulic fracturing have been widely recognized as the main technologies for shale oil production [8]. Although shale oil can be recovered at the initial stage after employing these developing methods, large amounts of shale resources are still left in shale reservoirs, up to 80% [9,10]. More advanced technologies should be developed to enhance shale oil recovery. There is a strong motivation for developing advanced extraction technologies in that even a 1.0% increase in oil recovery results in the production of an extra 1.6–9.0 billion barrels of oil [11].

Gas injection is accepted as being more advanced than traditional technologies for shale recovery. Gas is superior at entering the nanopores and therein has a higher sweep efficiency in shale reservoirs. Commonly used gases in oilfields are N_2_, CO_2_, natural gas, etc., of which CO_2_ possesses the lowest minimum miscibility pressure (MMP) compared to other gases [12,13]. The basic mechanism of CO_2_ for oil recovery includes oil swelling and considerable oil viscosity reduction [14,15,16,17,18]. When injected, CO_2_ can become miscible with the crude oil when the pressure is higher than the MMP. However, the miscible conditions are not easy to achieve in shale reservoirs. In fact, the light and intermediate hydrocarbons are readily recovered by CO_2_, while the heavy components remain in nanopores and are hard to recover [19].

Recently, methane, propane, butane, etc., have been proposed as cosolvents to assist CO_2_ for oil recovery, through which the performance of CO_2_ is reinforced [20,21,22,23]. With the assistance of a cosolvent, the miscibility performance of CO_2_ is greatly improved in the in situ oil; in addition, a cosolvent can also dissolve and reduce the viscosity of crude oil, which is beneficial for oil recovery. Methanol is used with CO_2_ to remove the liquid blockage in unconventional condensate reservoirs [24,25]. Liu et al. (2022) proposed dimethyl ether to assist CO_2_ flooding for oil recovery, as well as carbon sequestration, so-called storage-driven CO_2_ enhanced oil recovery (EOR) [26,27]. In order to improve shale oil recovery, more advanced technologies need to be developed. Hydrochloric saline lake shale formations are dominated by inorganic pores and have a low incidence of organic pores [28], which is of great significance for the design of CO_2_ injection wells. In order to improve the design of CO_2_ injection, it is critical to have a basic understanding of the fundamental mechanisms of shale hydrocarbon recovery during CO_2_ injection [29,30,31,32,33,34,35,36,37,38,39]. Advanced heating technologies like radio frequency and microwave heating can enhance carbon dioxide removal and recovery, improving energy efficiency and environmental sustainability. J.F. et al. demonstrated that radiofrequency(RF)-heated fixed-bed reactors utilizing CaCO_3_ sorbent can enhance CO_2_ capture from flue gas [40], while T.C. et al. found that Microwave Swing Desorption (MSD) accelerates CO_2_ desorption from microporous activated carbon fourfold compared to Temperature Swing Desorption (TSD) [41], together, showcasing innovative approaches to improving CO_2_ management and emissions reduction.

In this work, the Grand Canonical Monte Carlo (GCMC) simulation method is employed to investigate the adsorption/desorption behavior of CH_4_-C_4_H_10_ and CH_4_-C_4_H_10_-CO_2_ mixtures in organic and inorganic nanopores during pressure drawdown and CO_2_ huff and puff processes. Specifically, the influence of pore size, mineral types of nanopores, and pressure on adsorption and desorption behavior are thoroughly evaluated. The fundamental mechanism of shale gas recovery using the CO_2_ injection method is thereby revealed from the nanopore-scale perspective. This study enhances the understanding of CO_2_ injection in shale oil and gas recovery, offering a theoretical foundation for optimizing extraction technologies. By examining adsorption and desorption behaviors in nanopores, the research improves recovery efficiency and promotes sustainability. Additionally, the findings contribute to emission reduction, carbon sequestration, and the global energy crisis, supporting energy transition. Overall, the research holds significant academic value and potential applications, benefiting national energy security and environmental protection.

## 2. Molecular Models and Simulations

The mineral analysis indicates that Chang 7 shale is rich in organic kerogen and K-illite. The surface morphology of Chang 7 shale suggests that the shape morphology of the nanopores can be simplified as slit-shaped in the molecular models. The organic pore is usually represented by the carbon-based pore. In this molecular model, the carbon-slit pore is employed to represent the kerogen nanopore for simplicity, i.e., organic pores; the K-illite nanopore is used to represent the inorganic nanopore. In this study, the inorganic characteristics of the pores are predominantly influenced by the structural features of K-illite, known for its stable crystal framework and high thermal resistance. Conversely, the organic pores are derived from the organic kerogen-rich Chang 7 shale, which demonstrates excellent adsorption capacity and chemical reactivity. Consequently, both the inorganic and organic pores possess distinct advantages in terms of functionality and properties, collectively impacting the physicochemical performance of the shale. It should be noted that the K-illite has one Al-O octahedral sheet lying between each of the two Si-O tetrahedral sheets, i.e., one typical 2:1 clay mineral for Chang 7 shale [42]. The K-illite has the unit cell formula K_x_[Al_x_Si_(8-x)_][Al_y_Mg_4-y_]O_20_(OH)_4_ [43], in which the *x* and *y* represent 1 and 4, respectively. The clay sheet is negatively charged because one Si^4+^ cation is usually replaced by one Al^3+^ cation in every unit cell. Meanwhile, in the interlayers, the K^+^ cations are distributed randomly, counterbalancing the negative charges in the unit cell. It is noted that we investigate the recovery mechanism of shale gas in organic and inorganic nanopores with two sizes, i.e., 1 nm and 3 nm, representing micropores and mesopores, respectively.

The force field is used for the potential models of adsorbed gas [44]. The clay models are obtained from the CLAYFF force field. The C and H atoms in the alkanes are regarded as a united atom, while the partial charge of the C atom and O atoms in the CO_2_ molecule are taken as +0.7e and −0.35e, respectively. The following potential model is used to describe the Lennard-Jones (LJ) potential and the electrostatic terms for the nonbonded interactions:(1)$$u\left(r_{ij}\right)=4\varepsilon_{ij}\left[{\left(\frac{\sigma_{ij}}{r_{ij}}\right)}^{12}-{\left(\frac{\sigma_{ij}}{r_{ij}}\right)}^{6}\right]+\frac{q_{i}q_{j}}{4\pi \varepsilon_{0}r_{ij}}$$
where $\varepsilon_{ij}$ is the well depth of the *LJ* potential, and the $\sigma_{ij}$ is the *LJ* radius; *r_ij_* is the distance between atoms *i* and *j*; and *q* represents the atrial atom charge, which is employed to calculate the Coulomb interactions.

The cross interactions between two different atoms are calculated by the Lorentz–Berthelot combining rules as follows [45]:(2)$$\sigma_{ij}=\frac{1}{2}\left(\sigma_{ii}+\sigma_{jj}\right)$$
(3)$$\varepsilon_{ij}=\sqrt{\varepsilon_{ii}\varepsilon_{jj}}$$

The Ewald summation method is used to estimate the long-range electrostatic interactions by placing a vacuum slab in the *z* direction in the simulation cell [46,47].

In this study, the adsorption and desorption behavior of alkane/CO_2_ mixtures is investigated by the GCMC simulation method. The grand canonical *μVT* ensemble is employed in this work, where *μ*, *V*, and *T* represent the chemical potentials, volume, and temperature, respectively; within such an ensemble, *μ*, *V*, and *T* are regarded as constants [48]. Within the molecular simulation, gas molecules in nanopores can exchange with a fictitious bulk reservoir with a fixed chemical potential. At the beginning of constructing the GCMC simulation, we set the starting point of the simulation by selecting a certain number of C_4_H_10_ and CO_2_ molecules and randomly distributing them in the nanopores to represent the initial state. This initial configuration undergoes multiple cycles of energy minimization to ensure the thermodynamic stability of the system. A configurational-biased GCMC algorithm is used to insert and remove C_4_H_10_ molecules from nanopores [48]. The chemical potentials of bulk CH_4_ and C_4_H_10_ molecules in the canonical ensemble are calculated using Widom’s insertion method [49]. During GCMC simulations, the nanopore structure is fixed. The transitional move is implemented with CH_4_ molecules, which removes them from or inserts them into the simulation box randomly, with equal probability. As for CO_2_ and C_4_H_10_, in addition to the transitional move, a rotational move is implemented [48]. The Peng–Robinson equation of state (PR-EOS) is used to calculate the bulk densities [50]. The gas molecules are endowed with an equal probability; according to the chemical potential of the bulk reservoir, the gas molecules can be inserted or removed from the nanopores. The bulk reservoir has a given pressure and temperature. In order to initiate gas production, the bulk pressure is lowered from the initial pressure (P_i_ = 400 bar) to P_1_ = 200 bar and P_3_ = 100 bar. The equilibrium pressure (P_2_) in the nanopores after gas injection is calculated using the PR-EOS [50]. The equilibrium fluid composition is calculated based on the chemical potentials in and out of the nanopores.

As for the depressurization and CO_2_ huff-n-puff processes performed by GCMC, the primary pressure drawdown is first performed to initiate shale hydrocarbon production by lowering the bulk pressure from P_i_ = 400 bar to P_1_ = 200 bar. Subsequently, CO_2_ is introduced into the bulk volume. The PR-EOS is used to calculate the equilibrium pressure after CO_2_ injection, i.e., P_2_ [50]. After reaching equilibrium, the puffing process, i.e., a pressure drawdown process, is applied, starting from P_2_ to P_3_ = 100 bar. In order to achieve equilibrium, 0.3 million cycles are required for each molecule and 0.7 million cycles are required for sampling. This timescale ensures the stability and accuracy of the simulation process.

Through statistical analysis and comparison of the simulation results, we validated the effectiveness of the GCMC simulation method, which provides a solid theoretical foundation and an experimental basis for a deeper understanding of the adsorption and desorption behaviors of alkane/CO_2_ mixtures.

## 3. Results and Discussion

### 3.1. Primary Production during Depressurization

#### 3.1.1. Organic Nanopores

Figure 1, Figure 2, Figure 3 and Figure 4 present the density distribution of CH_4_, C_4_H_10_, and CH_4_-C_4_H_10_ mixtures in the micropores and mesopores at P_i_ and P_1_, respectively. It is noted that carbon slit pores were employed to represent the organic pores in shale, considering that carbon slit has similar physical properties to that of the organic matter in shale. In the micropores and mesopores, the density of hydrocarbons near the surface was significantly higher than that in the pore center, indicating that hydrocarbons form adsorption layers on the pore surface, while hydrocarbon molecules tended to stay in a free-gas state due to the relatively weak surface attraction. In addition, the density of hydrocarbons in the adsorption layers was remarkably higher than that in the mesopores due to the coupling effect of surface attraction from both sides of the nanopores. The density at the pore center of mesopores approached the bulk density, indicating that the gas readily formed free gas in the mesopores. However, the density at the pore center of the micropores did not converge with the bulk density, suggesting that there was no free-gas region.

As shown in Figure 1 and Figure 2, the density of CH_4_ decreases both in the adsorption layers and at the center of nanopores as the pressure declines. However, there is no obvious density change for C_4_H_10_ whether in the micropores or in the mesopores. In addition, for CH_4_-C_4_H_10_ mixtures, the density of CH_4_ drops in the first adsorption layer as pressure decreases, while C_4_H_10_ shows enrichment near the pore surface during pressure drawdown, indicated by increasing density at a lower pressure. During the primary gas production, the lighter hydrocarbons are more likely desorbed from the pore surface and more easily produced as the reservoir pressure drops, like CH_4_. On the contrary, due to the stronger interaction with the pore surface, the heavier components tend to be trapped in these organic nanopores, especially in micropores. This behavior is similar to that found in other adsorbents, such as zeolites [51] and graphene [52,53].

#### 3.1.2. Inorganic Nanopores

Figure 5, Figure 6, Figure 7 and Figure 8 present the density distribution of CH_4_ and C_4_H_10_ in a CH_4_-C_4_H_10_ mixture in the inorganic micropores and mesopores at P_i_ and P_1_, respectively. In this work, the inorganic nanopores are represented by K-illite pores. Similarly to that in the organic pores, CH_4_ and C_4_H_10_ molecules form an adsorption layer near the pore surface and free-gas region at the pore center in mesopores, while there is no free gas in the micropores. Comparatively, the density in the adsorption layers in the inorganic pores is much smaller than that in the organic pores, demonstrating relatively weaker molecule–surface interactions in the inorganic pores. As the pressure drops, the density of CH_4_ in the adsorption layer decreases, while the adsorption layer of C_4_H_10_ cannot be recovered during pressure drawdown. Specifically, as with the CH_4_-C_4_H_10_ mixture, the density of C_4_H_10_ increases in the first adsorption layer, and the increase is remarkably more significant in the inorganic pores than that of the organic nanopores. In other words, it is more difficult to recover the heavier hydrocarbons from the inorganic pores, such as C_4_H_10_.

### 3.2. Improved Gas Production during CO_2_ Huff-n-Puff Process

#### 3.2.1. Organic Nanopores

In this process, CO_2_ is injected into the micro- and mesopores; the system pressure is increased from P_1_ to P_2_ during the huffing process and then it decreases to P_3_ in the puffing process. Figure 9, Figure 10, Figure 11 and Figure 12 present the density profiles of CH_4_ and C_4_H_10_ in the CH_4_-C_4_H_10_ mixture in the organic micropores and mesopores at P_1_, P_2_, and P_3_, respectively. Comparatively, CO_2_ presents the strongest adsorption capacity on the pore surface compared to CH_4_ and C_4_H_10_. In addition, it is found that the adsorption layer of CO_2_ is much closer to the pore surface than those of the alkanes, i.e., CH_4_ and C_4_H_10_. After introducing CO_2_, i.e., in the CO_2_ huffing period (P_1_ → P_2_), the density in the first adsorption layer of CH_4_ and C_4_H_10_ decreases drastically, indicating that the adsorbed hydrocarbons are replaced by the injected CO_2_. This observation agrees well with the previous work [19,54]. In the puffing process, i.e., P_2_ → P_3_, the adsorbed CH_4_ desorbs from the pore surface continuously and thus can be heavily recovered. However, the density of C_4_H_10_ in the adsorption layer is slightly changed, suggesting that the adsorbed C_4_H_10_ cannot be readily produced. More interestingly, as puffing time continues, the adsorbed CH_4_ in the CH_4_-C_4_H_10_ mixture is produced as expected, while the density of C_4_H_10_ in the adsorption layer increases, indicating that C_4_H_10_ re-adsorbs on the pore surface in the huffing period.

#### 3.2.2. Inorganic Nanopores

Figure 13, Figure 14, Figure 15 and Figure 16 present the density profiles of CH_4_, C_4_H_10_, and the CH_4_-C_4_H_10_ mixture in the inorganic micropores and mesopores at P_1_, P_2_, and P_3_, respectively. Unlike in organic pores, the density of CO_2_ in the adsorption layer within inorganic pores is significantly higher. It was reported that CO_2_ molecules have a neutral charge and zero dipole moment, while CO_2_ molecules exhibit a strong quadrupole moment [55,56]. These properties enable CO_2_ to have improved adsorption by combining with the charged clay atoms [46]. Due to the presence of CO_2_ adsorption layers, CH_4_ and C_4_H_10_ form a second minor adsorption layer towards the pore center. In the inorganic mesopores, the density of CH_4_ and C_4_H_10_ decreases significantly in the adsorption layer during the CO_2_ huffing period. That is, the adsorbed CH_4_ and C_4_H_10_ can be recovered efficiently from the inorganic mesopores during the CO_2_ huffing period. On the contrary, in the inorganic micropores, the density of C_4_H_10_ in the adsorption layer slightly increases during the CO_2_ puffing period.

### 3.3. Shale Hydrocarbon Recovery

The oil recovery efficiency (*RE*) is obtained for CH_4_ and C_4_H_10_ from the organic and inorganic micro- and mesopores. *RE* is calculated by $\int_{z_{0}}^{z_{i}}\rho_{P_{i}}dv/\int_{z_{0}}^{{z^{\prime }}_{0}}\rho_{P_{0}}dv$ (where $\rho_{P_{i}}$ and $\rho_{P_{0}}$ represent the density profiles at *P_i_* and *P*_0_). Figure 17, Figure 18, Figure 19 and Figure 20 present the recovery efficiency during pressure drawdown and the CO_2_ huffing and puffing period. We find that RE during pressure drawdown is smaller in micropores than in mesopores, and vice versa for the heavier hydrocarbons, i.e., C_4_H_10_. In inorganic pores, the recovery efficiency of C_4_H_10_ is much higher than that of organic pores. However, RF is comparable in both kinds of pores for the lighter hydrocarbons, i.e., CH_4_. Interestingly, the recovery efficiency for C_4_H_10_ can be negative in the micropores during pressure drawdown; in other words, the heavy hydrocarbons tend to accumulate in the small pores as pressure drops, which is in line with our previous findings. During CO_2_ huff and puff, the recovery efficiency of CH_4_ is higher in the organic pores than that of the inorganic pores; the efficiency during the CO_2_ huffing process is comparable in the micro- and mesopores. More importantly, we observe that the recovery efficiency of C_4_H_10_ reaches the highest in both the inorganic and organic pores during the CO_2_ huff and puff process. It suggests that the CO_2_ huff and puff method is more suitable for heavier hydrocarbon recovery compared to the pressure drawdown method.

### 3.4. CO_2_ Sequestration in Nanopores

Figure 21 and Figure 22 present the molar fraction of the free-state and adsorbed-state CO_2_ in the micro- and meso-organic and -inorganic pores, respectively. In the organic pores, CO_2_ is mainly sequestrated in the adsorbed state in micropores, while, in the mesopores, free-state CO_2_ dominates at low pressures. In large pores, most CO_2_ molecules are adsorbed on the pore surface during high-pressure conditions; however, as pressure decreases, the adsorbed CO_2_ desorbs from the pore surface and is sequestrated in the free state. Comparatively, in the small pores, CO_2_ molecules are mainly in adsorbed state storage due to the strong fluid–pore surface interactions. In the inorganic pores, CO_2_ is stored mainly in the free state due to the relatively weak fluid–pore surface interactions, except in the micropores at high-pressure conditions. Similarly, as pressure decreases, CO_2_ sequestration transfers from the adsorbed state to the free-gas state in the inorganic pores. In summary, CO_2_ sequestration in the adsorbed state is safer than that in the free state. A high content of organic matter, high pressure, and small pores are beneficial factors for transforming CO_2_ sequestration into adsorbed state storage.

## 4. Conclusions

In this work, the GCMC simulation methods were applied to study the mechanisms of shale hydrocarbon extraction in organic and inorganic nanopores with sizes of 1 nm and 3 nm, analyzing the impact of pressure drawdown and the CO_2_ huff and puff process on gas recovery rates. During the primary gas production, CH_4_ readily desorbs from the pore surface and is produced as reservoir pressure drops. On the contrary, C_4_H_10_ tends to be trapped in these organic nanopores and is difficult to produce, especially in the micropores. Comparatively, it is more difficult to recover C_4_H_10_ from the inorganic pores. During the CO_2_ puffing process, the adsorbed CH_4_ desorbs from the pore surface continuously and can therein be heavily extracted. However, the adsorbed C_4_H_10_ cannot be readily produced. During the CO_2_ huffing period, the adsorbed CH_4_ and C_4_H_10_ can desorb and escape from the inorganic mesopores, enhancing shale gas production. However, during the CO_2_ puffing period, C_4_H_10_ in the inorganic micropores is hard to recover. Research findings indicate that during pressure drawdown and the CO_2_ huff and puff process, CH_4_ is more easily desorbed and released from the pores, whereas C_4_H_10_ is more difficult to generate and recover, especially in inorganic micropores.

Recovery efficiency during pressure drawdown is smaller in micropores than that of the mesopores. In inorganic pores, the recovery efficiency of C_4_H_10_ is much higher than that in the organic pores. However, it is comparable in both pores for CH_4_. In addition, the recovery efficiency of C_4_H_10_ can be negative in the micropores during pressure drawdown. During CO_2_ huff and puff, the recovery efficiency of CH_4_ is higher in the organic pores than that in the inorganic pores. More importantly, the recovery efficiency of C_4_H_10_ reaches the highest for both the inorganic and organic pores during the CO_2_ huff and puff process. These findings indicate that the CO_2_ huff and puff method is more effective for recovering heavier hydrocarbons than the pressure drawdown method.

Beyond traditional carbon sequestration methods, CO_2_ mainly sequestrates in the adsorbed state in organic pores, while it mainly sequestrates in the free-gas state in the inorganic pores. According to the content of this study, CO_2_ sequestration in the adsorbed state is safer than that in the free state. In our work, it is found that a high content of organic matter, high pressure, and small pores are beneficial factors for CO_2_ sequestration transforming into adsorbed state storage.

## Figures and Tables

**Figure 1 nanomaterials-14-01698-f001:**
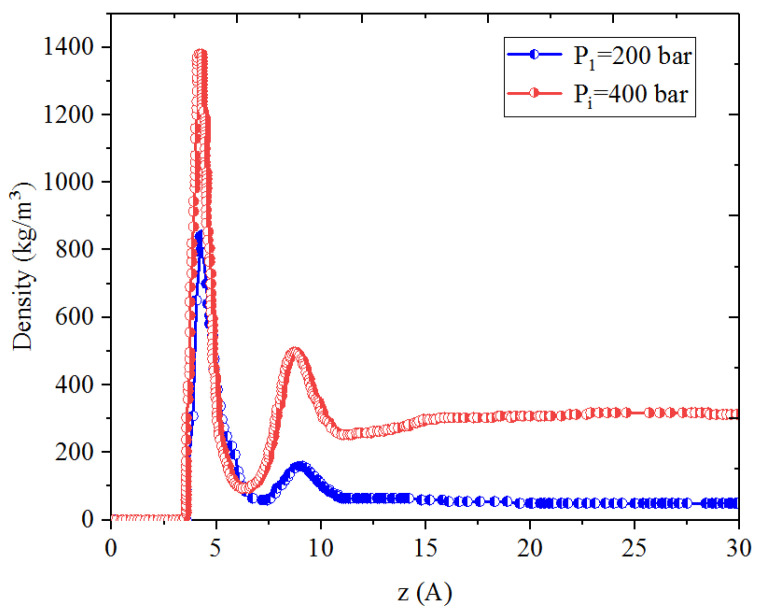
Density profiles of CH_4_ in CH_4_-C_4_H_10_ mixture in organic mesopores at 353.15 K.

**Figure 2 nanomaterials-14-01698-f002:**
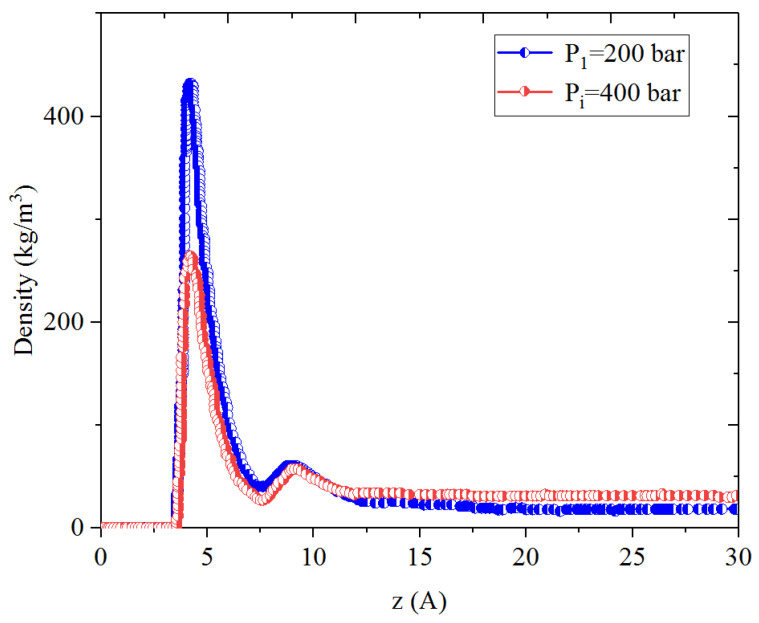
Density profiles of C_4_H_10_ in CH_4_-C_4_H_10_ mixture in organic mesopores at 353.15 K.

**Figure 3 nanomaterials-14-01698-f003:**
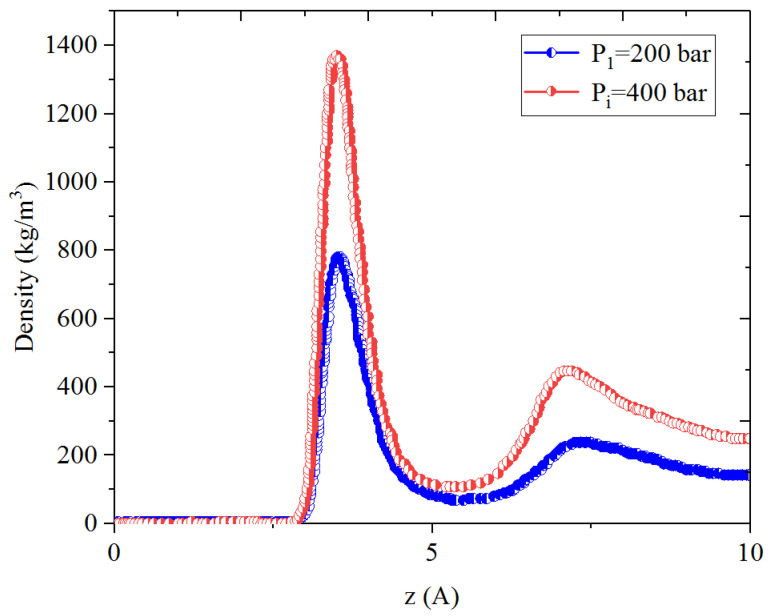
Density profiles of CH_4_ in CH_4_-C_4_H_10_ mixture in organic micropores at 353.15 K.

**Figure 4 nanomaterials-14-01698-f004:**
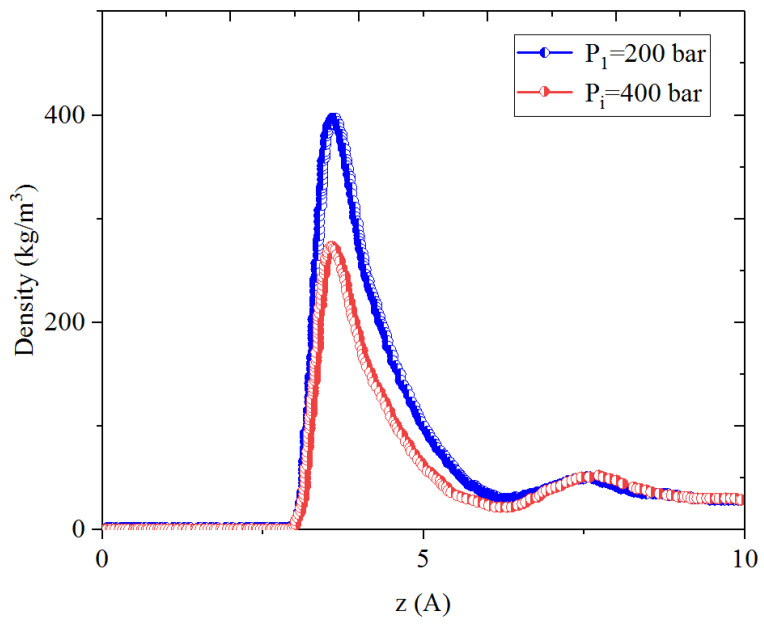
Density profiles of C_4_H_10_ in CH_4_-C_4_H_10_ mixture in organic micropores at 353.15 K.

**Figure 5 nanomaterials-14-01698-f005:**
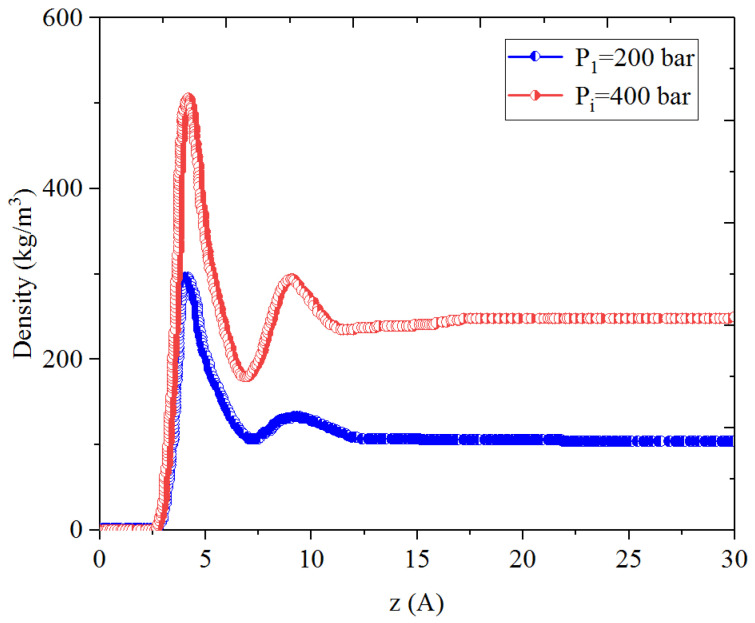
Density profiles of CH_4_ in CH_4_-C_4_H_10_ mixture in inorganic mesopores at 353.15 K.

**Figure 6 nanomaterials-14-01698-f006:**
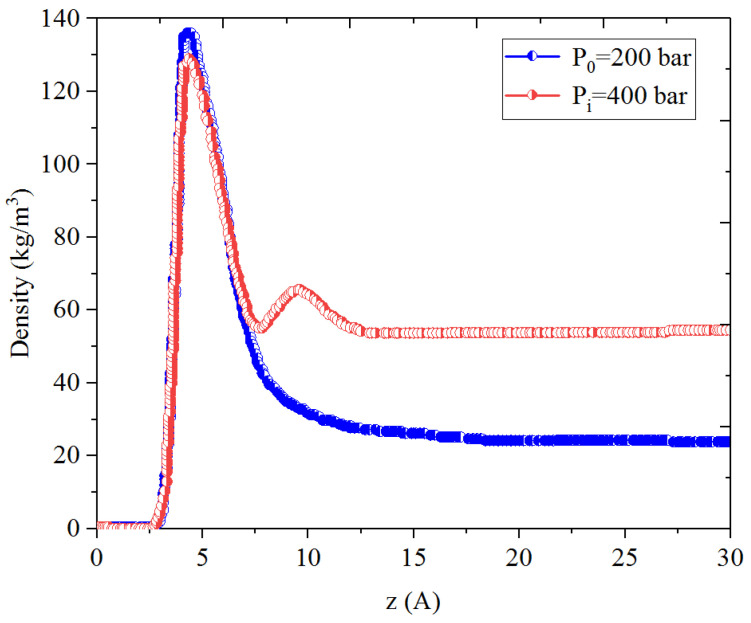
Density profiles of C_4_H_10_ in CH_4_-C_4_H_10_ mixture in inorganic mesopores at 353.15 K.

**Figure 7 nanomaterials-14-01698-f007:**
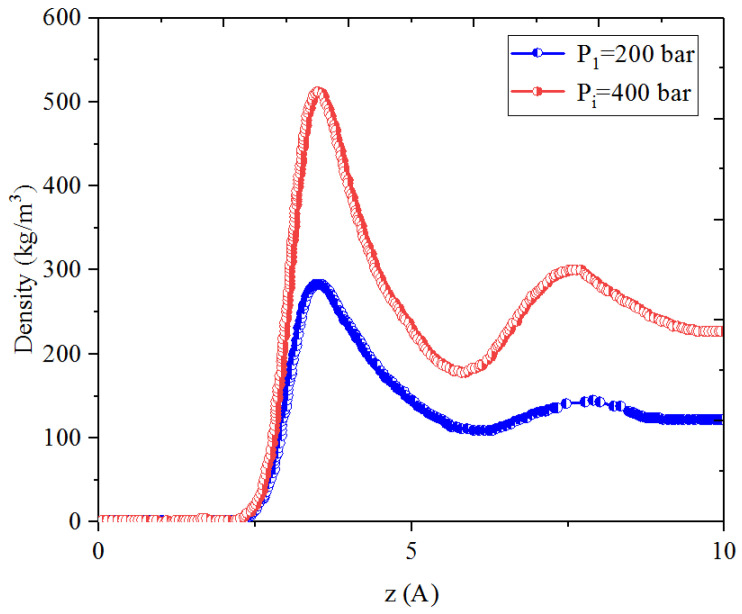
Density profiles of CH_4_ in CH_4_-C_4_H_10_ mixture in inorganic micropores at 353.15 K.

**Figure 8 nanomaterials-14-01698-f008:**
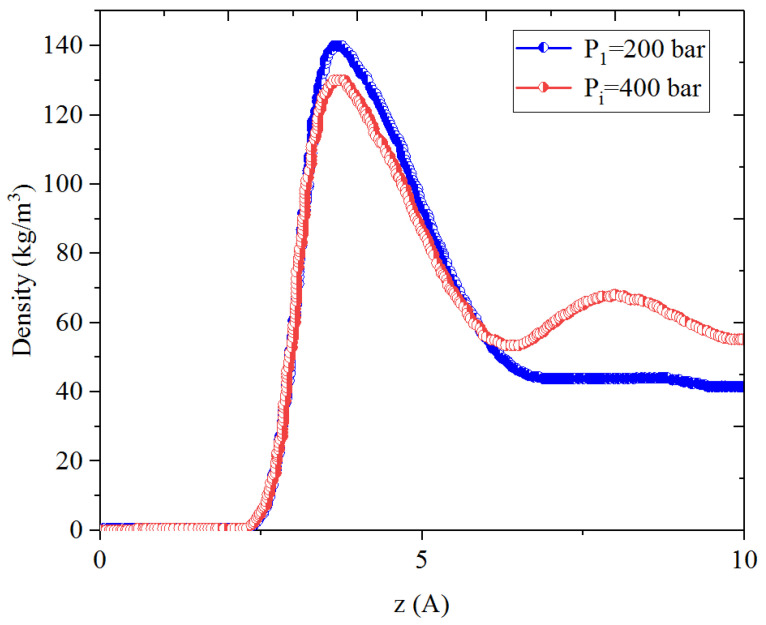
Density profiles of C_4_H_10_ in CH_4_-C_4_H_10_ mixture in inorganic micropores at 353.15 K.

**Figure 9 nanomaterials-14-01698-f009:**
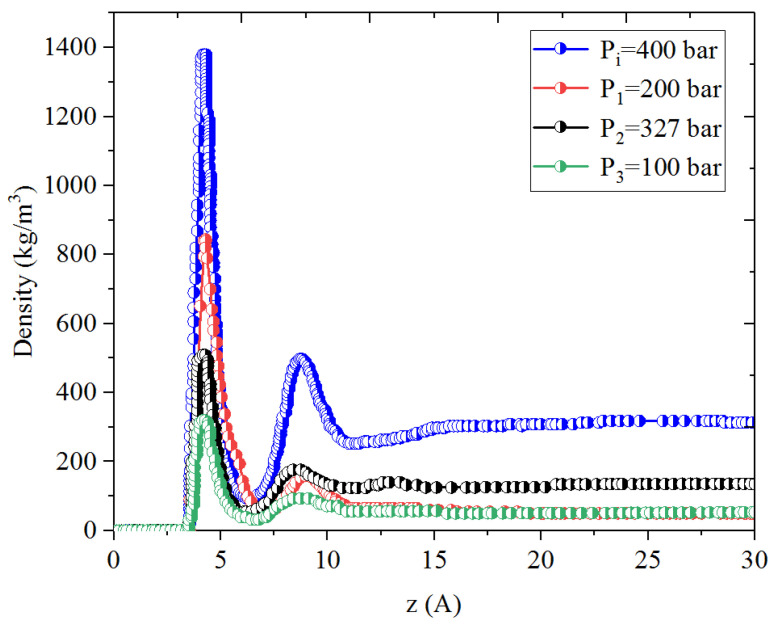
Density profiles of CH_4_ in CH_4_-C_4_H_10_-CO_2_ mixture in organic mesopores at P_i_, P_1_, P_2_, and P_3_, respectively.

**Figure 10 nanomaterials-14-01698-f010:**
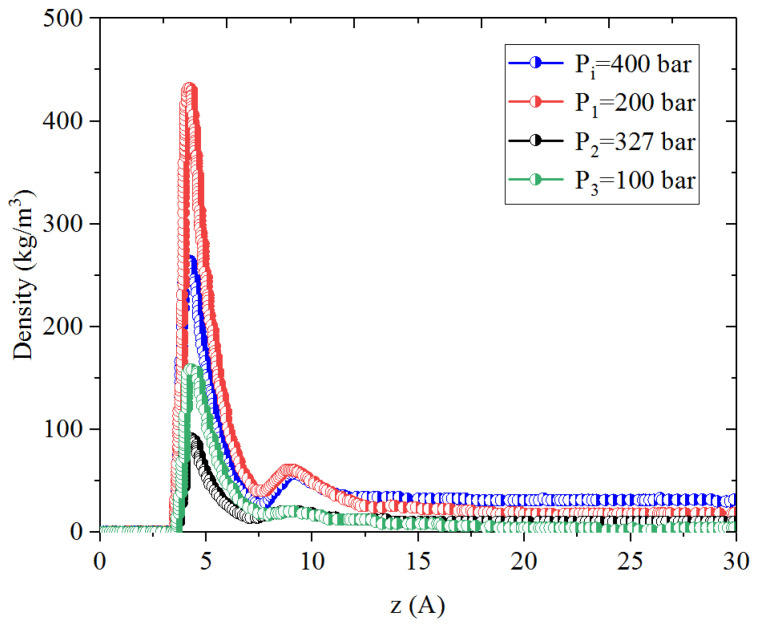
Density profiles of C_4_H_10_ in CH_4_-C_4_H_10_-CO_2_ mixture in organic mesopores at P_i_, P_1_, P_2_, and P_3_, respectively.

**Figure 11 nanomaterials-14-01698-f011:**
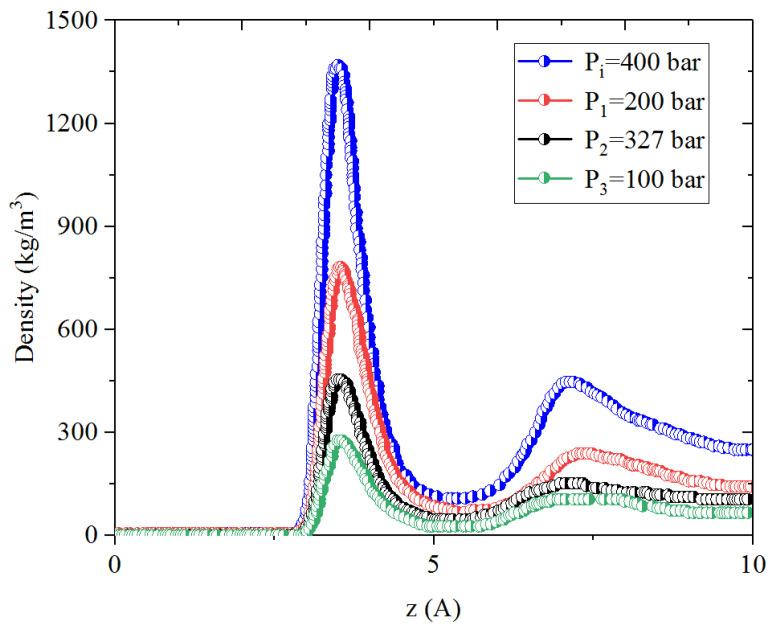
Density profiles of CH_4_ in CH_4_-C_4_H_10_-CO_2_ mixture in organic micropores at P_i_, P_1_, P_2_, and P_3_, respectively.

**Figure 12 nanomaterials-14-01698-f012:**
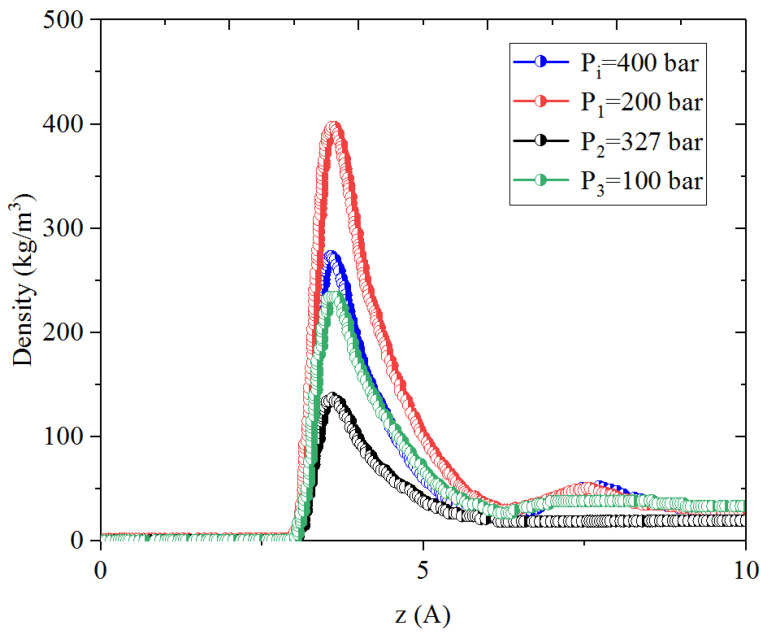
Density profiles of C_4_H_10_ in CH_4_-C_4_H_10_-CO_2_ mixture in organic micropores at P_i_, P_1_, P_2_, and P_3_, respectively.

**Figure 13 nanomaterials-14-01698-f013:**
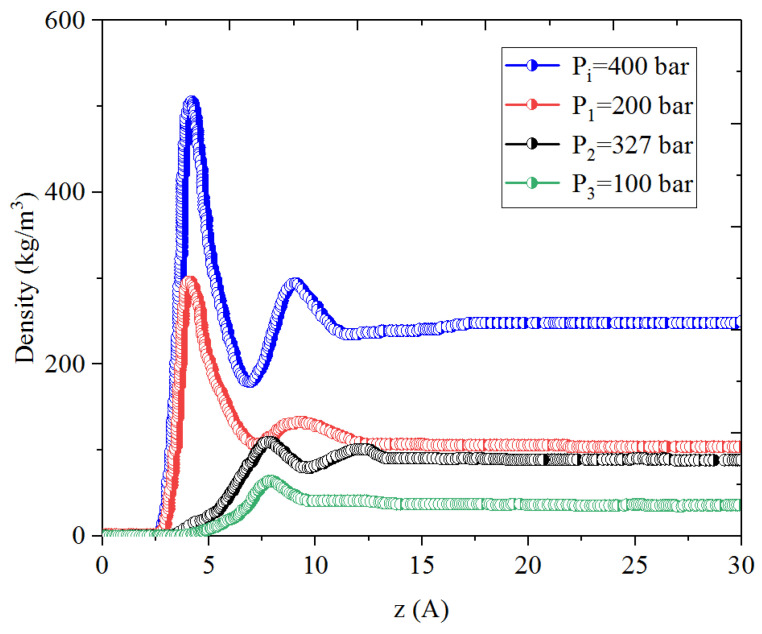
Density profiles of CH_4_ in CH_4_-C_4_H_10_-CO_2_ mixture in inorganic mesopores at P_i_, P_1_, P_2_, and P_3_, respectively.

**Figure 14 nanomaterials-14-01698-f014:**
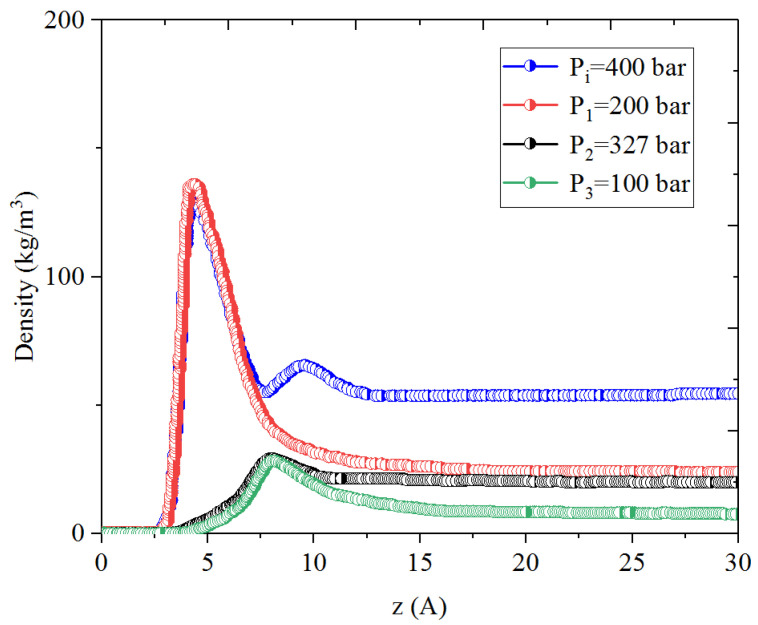
Density profiles of C_4_H_10_ in CH_4_-C_4_H_10_-CO_2_ mixture in inorganic mesopores at P_i_, P_1_, P_2_, and P_3_, respectively.

**Figure 15 nanomaterials-14-01698-f015:**
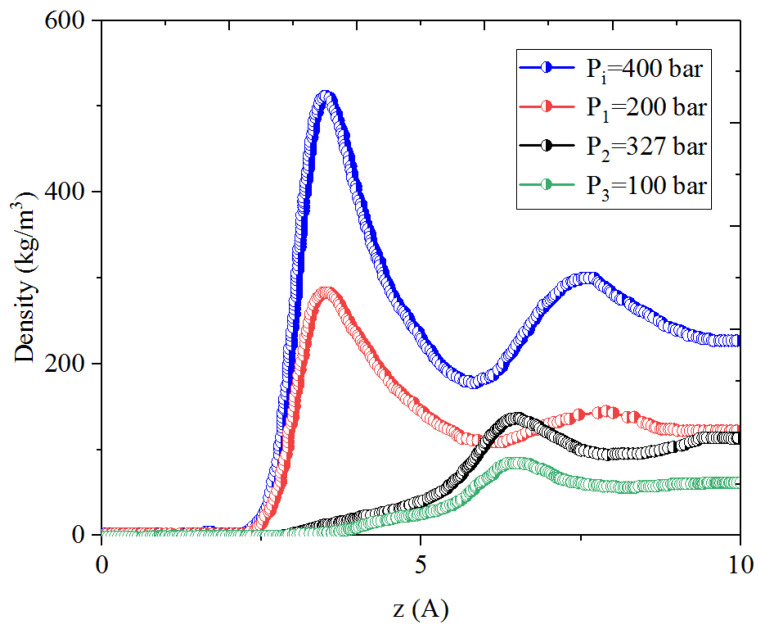
Density profiles of CH_4_ in CH_4_-C_4_H_10_-CO_2_ mixture in inorganic micropores at P_i_, P_1_, P_2_, and P_3_, respectively.

**Figure 16 nanomaterials-14-01698-f016:**
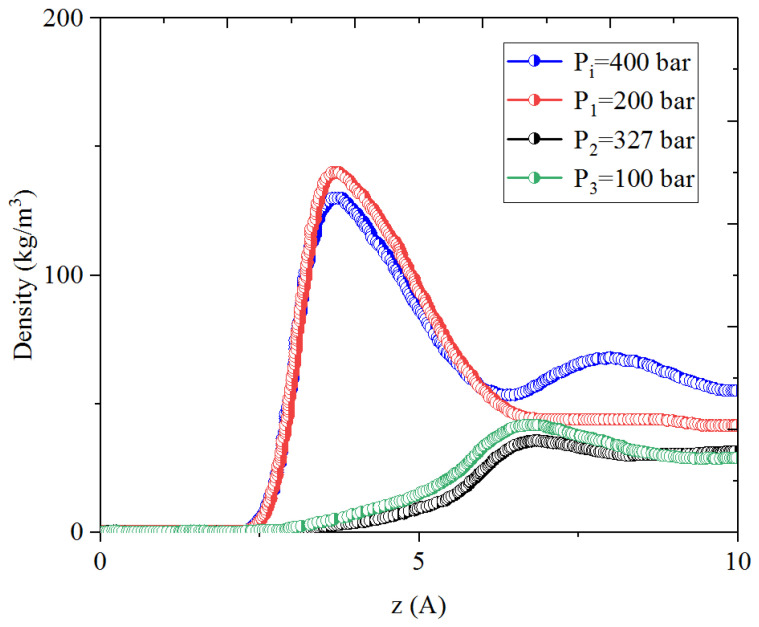
Density profiles of C_4_H_10_ in CH_4_-C_4_H_10_-CO_2_ mixture in inorganic micropores at P_i_, P_1_, P_2_, and P_3_, respectively.

**Figure 17 nanomaterials-14-01698-f017:**
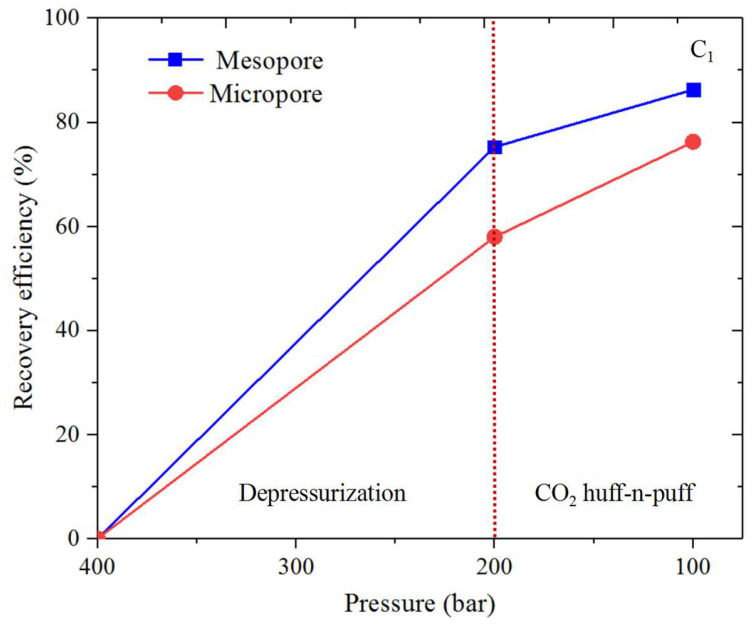
Recovery efficiency of CH_4_ from the organic nanopores during pressure drawdown and the CO_2_ huff and puff process.

**Figure 18 nanomaterials-14-01698-f018:**
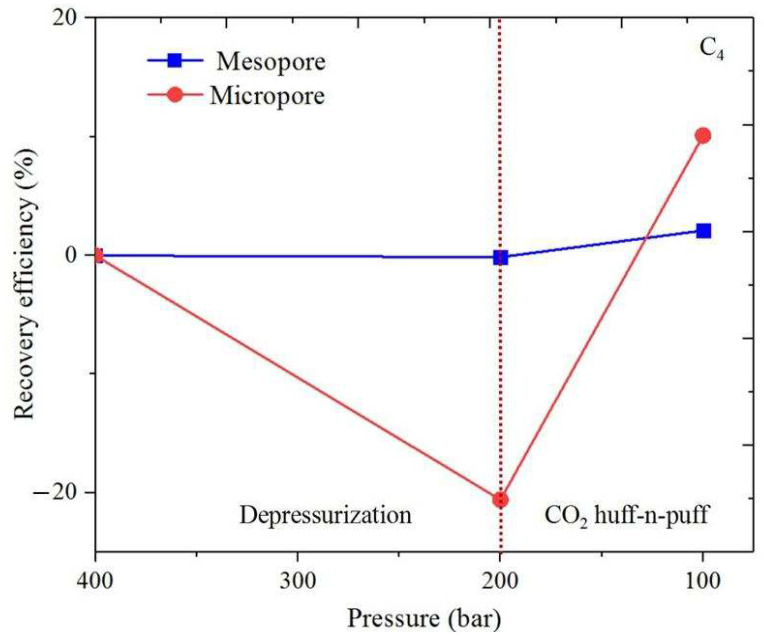
Recovery efficiency of C_4_H_10_ from the organic nanopores during pressure drawdown and the CO_2_ huff and puff process.

**Figure 19 nanomaterials-14-01698-f019:**
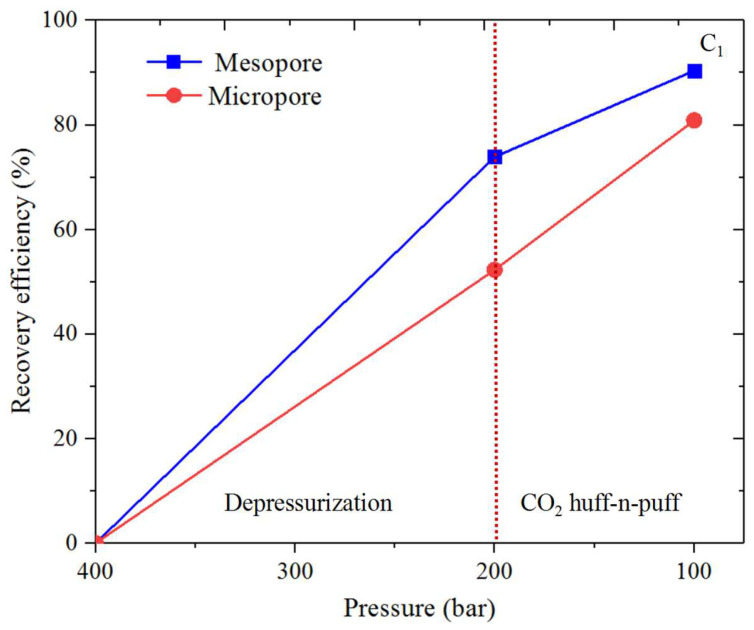
Recovery efficiency of CH_4_ from the inorganic nanopores during pressure drawdown and the CO_2_ huff and puff process.

**Figure 20 nanomaterials-14-01698-f020:**
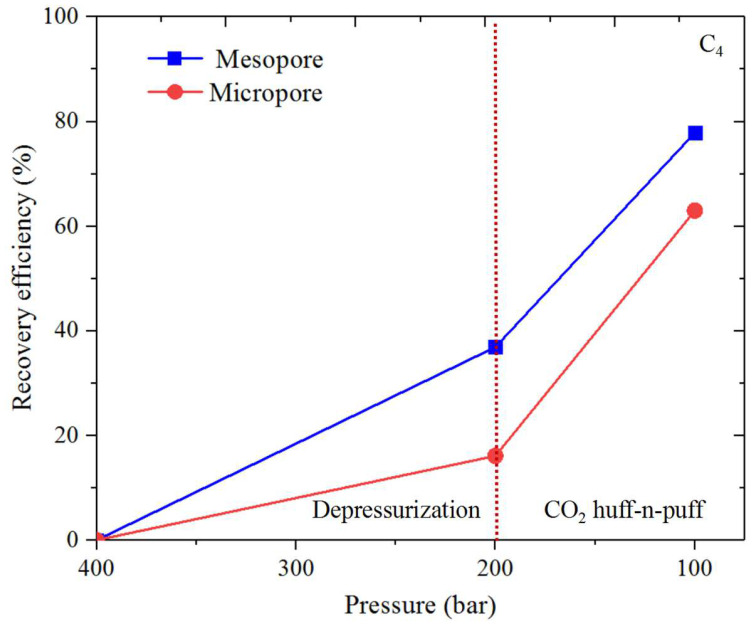
Recovery efficiency of C_4_H_10_ from the inorganic nanopores during pressure drawdown and the CO_2_ huff and puff process.

**Figure 21 nanomaterials-14-01698-f021:**
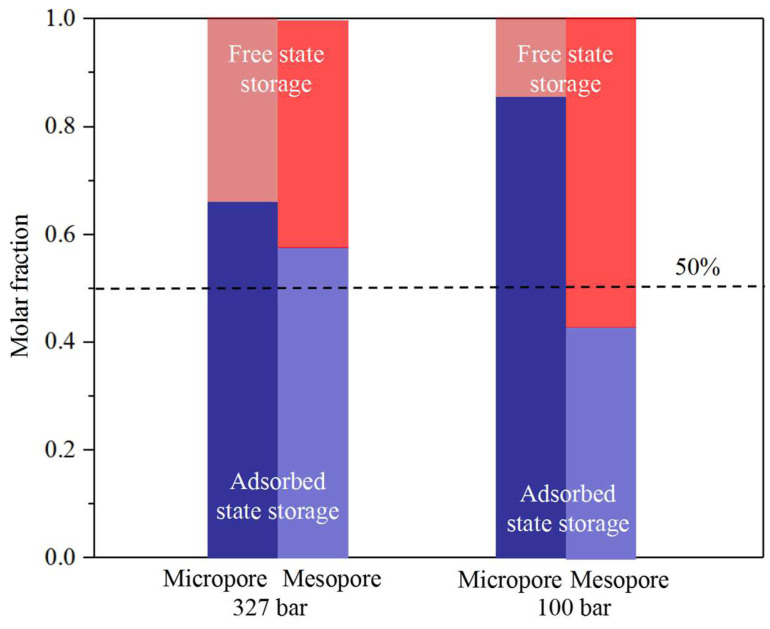
The molar fraction of the free-state and adsorbed-state CO_2_ in the micro- and meso-organic pores.

**Figure 22 nanomaterials-14-01698-f022:**
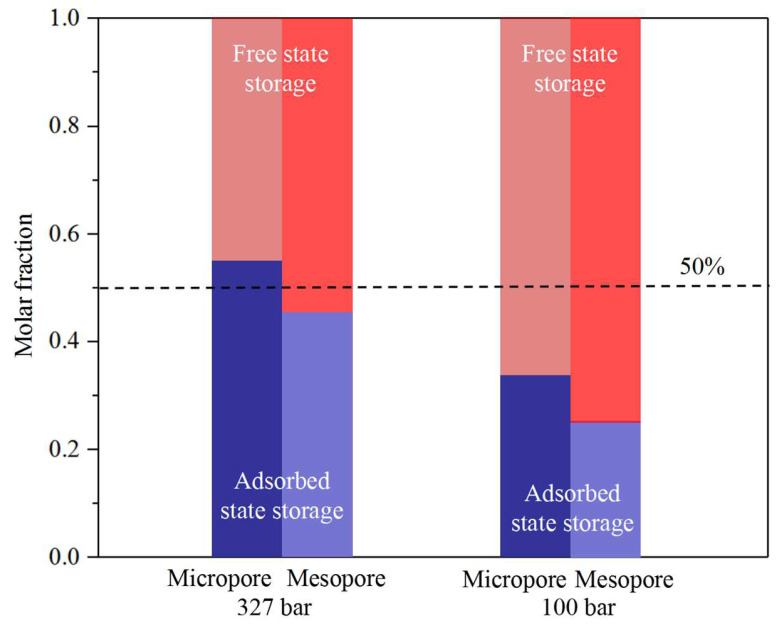
The molar fraction of the free-state and adsorbed-state CO_2_ in the micro- and meso-inorganic pores.

## Data Availability

We do not accept the public sharing of raw data. The raw data discussed in this study is included in the article materials, and further inquiries can be directed to the corresponding author.

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
