# Peer review of "CO2 Utilization and Sequestration in Organic and Inorganic Nanopores During Depressurization and Huff-n-Puff Process"

_nanomaterials, 2024, doi:10.3390/nano14211698_

Round 1

Reviewer 1 Report

Comments and Suggestions for Authors

The paper addresses CO2 Utilization and Sequestration in Organic and Inorganic Nanopores during Depressurization and Huff-n-Puff Process.

The research gap is clear and the authors study mechanism of shale gas recovery using the CO2 injection method is thereby revealed from the nanopore scale perspective, but the need and application for this paper is not clearly stated at the end of the introduction.

More references must be included in the introduction in order to compare other alternatives with the current selected technology:

https://doi.org/10.1016/j.cep.2016.07.004

https://doi.org/10.1016/j.micromeso.2014.06.032

Methodology: Starting points and validation of the simulation must be included in this section.

Results: Figure 18 shows a negative efficiency which does not make sense.

Conclusions do not include any main findings and specific values for pore size or conditions.

Comments on the Quality of English Language

English must be checked.

Author Response

Comments 1:[The research gap is clear and the authors study mechanism of shale gas recovery using the CO2 injection method is thereby revealed from the nanopore scale perspective, but the need and application for this paper is not clearly stated at the end of the introduction.]

Response 1: [This study enhances the understanding of CO2 injection in shale oil and gas recovery, offering a theoretical foundation for optimizing extraction technologies. By examining adsorption and desorption behaviors in nanopores, the research improves recovery efficiency and promotes sustainability. Additionally, the findings contribute to emission reduction, carbon sequestration, and the global energy crisis, supporting the energy transition. Overall, the research holds significant academic value and potential applications, benefiting national energy security and environmental protection.] Thank you for pointing this out. We agree with this comment. Therefore, [We have added the requirements and applications of the paper after the introduction.Mention exactly where in the revised manuscript this change can be found - page 3, paragraph 4, and line 7 to 14.

Comments 2:[More references must be included in the introduction in order to compare other alternatives with the current selected technology:

https://doi.org/10.1016/j.cep.2016.07.004

https://doi.org/10.1016/j.micromeso.2014.06.032]

Response 2: [ Advanced heating technologies like radio frequency and microwave heating can enhance carbon dioxide removal and recovery, improving energy efficiency and environmental sustainability. J.F. et al. demonstrated that RF heated fixed bed reactors utilizing CaCO3 sorbent can enhance CO2 capture from flue gas[40], while T.C. et al. found that Microwave Swing Desorption (MSD) accelerates CO2 desorption from microporous activated carbon fourfold compared to Temperature Swing Desorption (TSD)[41], together showcasing innovative approaches to improving CO2 management and emissions reduction.] Thank you for pointing this out. We agree with this comment. Therefore, [We have added the two references you provided to the introduction. Thank you for the references to help compare alternative solutions with the currently selected technology.Mention exactly where in the revised manuscript this change can be found - page 3, paragraph 3, and line 14 to 21.

Comments 3:[Methodology: Starting points and validation of the simulation must be included in this section.]

Response 3: [ At the beginning of constructing the GCMC simulation, we set the starting point of the simulation by selecting a certain number of C4H10 and CO2 molecules and randomly distributing them in the nanopores to represent the initial state. This initial configuration underwent multiple cycles of energy minimization to ensure the thermodynamic stability of the system. ]  [This time scale ensures the stability and accuracy of the simulation process.

Through statistical analysis and comparison of the simulation results, we validated the effectiveness of the GCMC simulation method, which provides a solid theoretical foundation and experimental basis for a deeper understanding of the adsorption and desorption behaviors of alkane/COâ‚‚ mixtures.] Thank you for pointing this out. We agree with this comment. Therefore, [We have incorporated the initial points of the simulation and the validation of the simulation into the article, thereby enhancing the overall methodology. Thank you for your suggestions.Mention exactly where in the revised manuscript this change can be found - Page 3, paragraph 6,line 6 to 8. Page 4, paragraph 1, line 1 to 3.Page 4, paragraph 2, line 8 to 9.Page 4, paragraph 3, line 1 to 4

Comments 4:[Results: Figure 18 shows a negative efficiency which does not make sense.]

Response 4: [Figure 18 shows that the recovery efficiency for C4H10 can be negative in the micropores during pressure drawdown. We have already prepared a graph that only displays positive efficiency. If you think this is inappropriate, we can change it at any time. Thank you for your suggestion.]

Comments 5:[Conclusions do not include any main findings and specific values for pore size or conditions.]

Response 5: [ In this work, the GCMC simulation methods were applied to study the mechanisms of shale hydrocarbon extraction in organic and inorganic nanopores with sizes of 1 nm and 3 nm, analyzing the impact of pressure drawdown and the CO2 huff and puff process on gas recovery rates. ]  [Research findings indicate that during pressure drawdown and CO2 huff and puff process, CH4 is more easily desorbed and released from the pores, whereas C4H10 is more difficult to generate and recover, especially in inorganic micropores.] [These findings indicate that the CO2 huff and puff method is more effective for recovering heavier hydrocarbons than the pressure drawdown method.] Thank you for pointing this out. We agree with this comment. Therefore, [We have incorporated the main findings and specific values of the aperture conditions in the conclusion.Mention exactly where in the revised manuscript this change can be found - Page 16, paragraph 1,line 1to 4. Page 16, paragraph 1, line 13 to 15.Page 16, paragraph 2, line 8 to 9.

Reviewer 2 Report

Comments and Suggestions for Authors

Author Response

Comments 1:[The manuscript has many positive qualities. It is data rich, and generally well-written and organized. I will  leave the technical analysis to reviewers with more expertise than me; my comments mostly refer to details of grammar and phrasing. The content refers to micropores, but I am not sure that this is an appropriate submission to Nanomaterials; perhaps Processes would be a better choice. The nanopore description is restricted to a manuscript supplement. This is a weakness for a manuscript being submitted to a journal focusing on nanomaterials. .]

Response 1: Thank you for pointing this out. [Micropores refer to pores with diameters smaller than 2 nanometers, while nanopores is a broader term that can include both micropores and mesopores. When discussing pore sizes, it is common to distinguish between micropores, mesopores, and nanopores based on specific size ranges. Therefore, I believe this is suitable for submission to Nanomaterials.]

Comments 2:[Throughout the manuscript, the authors refer inorganic and organic micropores. By definition, a micropore is simply a small void. The inorganic and organic nature of the pores needs to be explained.]

Response 2: [In this study, the inorganic characteristics of the pores are predominantly influenced by the structural features of K-illite, known for its stable crystal framework and high thermal resistance. Conversely, the organic pores are derived from the organic kerogen-rich Chang 7 shale, which demonstrates excellent adsorption capacity and chemical reactivity. Consequently, both the inorganic and organic pores possess distinct advantages in terms of functionality and properties, collectively impacting the physicochemical performance of the shale.] Thank you for pointing this out. We agree with this comment. Therefore, We have incorporated the inorganic and organic properties of the pores into the article.Mention exactly where in the revised manuscript this change can be found - page 3, paragraph 1, and line 7 to 13. 

Comments 3:[I suggest starting with a definition of the huff and puff process, instead of leaving readers to figure out the process from the context of subsequent statements about CO2]

Response 3: [The huff and puff process involves injecting CO2 into the micro- and mesopores, where the system pressure is increased during the huffing process and decreased during the puffing process. ] Thank you for pointing this out. We agree with this comment. Therefore, We have added the definitions to the summary.Mention exactly where in the revised manuscript this change can be found - page 1, paragraph 1, and line 5 to 7. 

Comments 4:[CH4 are more likely produced as the reservoir pressure drops. On the contrary, C4H10 are tending to be trapped Comment: Tense mismatch – should be CH4 is and C4H10 is]

Response 4: [CH4 is more likely produced as the reservoir pressure drops. On the contrary, C4H10 is .] Thank you for pointing this out. We agree with this comment. Therefore, [We have made the changes as requested.Mention exactly where in the revised manuscript this change can be found - page 1, paragraph 1, and line 9 to 10. 

Comments 5:[and are hard to be produced Comment: You are referring to extracting them, not producing them]

Response 5: [nanopores and are hard to be extracted,] Thank you for pointing this out. We agree with this comment. Therefore, We have made the changes as requested.Mention exactly where in the revised manuscript this change can be found - page 1, paragraph 1, and line 11 . 

Comments 6:[The strong motivation behind that is that it is estimated that Comment: More graceful wording would be The strong motivation for developing advanced extraction technologies is that even an]

Response 6: [The strong motivation for developing advanced extraction technologies is that even an 1.0% increase.] Thank you for pointing this out. We agree with this comment. Therefore, We have made the changes as requested.Mention exactly where in the revised manuscript this change can be found - page 2, paragraph 1, and line 5 to 6. 

Comments 7:[Recently, cosolvent, such as methane, propane, butane etc., is proposed to assist CO2 for oil recovery, Comment: Clearer wording would be Methane, propane, butane etc., are proposed as cosolvents to assist CO2 for oil recovery,]

Response 7: [Methane, propane, butane etc., are proposed as cosolvents to assist CO2 for oil recovery, ] Thank you for pointing this out. We agree with this comment. Therefore, We have made the changes as requested.Mention exactly where in the revised manuscript this change can be found - page 2, paragraph 3, and line 1 to 2. 

Comments 8:[Hydrochloric saline lake shale formations Comment: I am not sure what you mean by a “hydrochloric” saline lake shale. A saline lake containing NaCL , with an acidic pH?]

Response 8: [Hydrochloric saline lake shale formations refer to shale strata that are formed in saline lake environments, which may contain rich organic matter, minerals, and salts. These shales are usually formed in depositional environments of saline lakes, influenced by water bodies with higher concentrations of salinity.] Thank you for pointing this out.

Comments 9:[Note that the organic pore is usually represented by the carbon-based pore. In this molecular model, the carbon-slit pore is employed to represent the kerogen nanopore for simplicity, i.e., organic pores; the K-illite nanopore is used to represent the inorganic nanopore. Note that the K-illite has  Comment: These descriptions would be more gracious if the “Note that…” beginnings were eliminated. These commands to the reader serve no useful purpose.]

Response 9: [The organic pore is usually represented by the carbon-based pore. ] Thank you for pointing this out. We agree with this comment. Therefore, We have made the changes as requested.Mention exactly where in the revised manuscript this change can be found - page 3, paragraph 1, and line 4. 

Comments 10:[the Chang 7 shale is rich of organic kerogen and K-illite Comment: wording should be rich in, not rich of]

Response 10: [ the Chang 7 shale is rich in organic kerogen] Thank you for pointing this out. We agree with this comment. Therefore, [We have made the changes as requested.Mention exactly where in the revised manuscript this change can be found - page 3, paragraph 1, and line 1. 

Comments 11:[Based on the mineral analysis, the Chang 7 shale is rich of organic kerogen and Killite. Based on the surface morphology of Chang 7 shale, shape morphology of the nanopores is simplified as the slit-shaped in the molecular modelsComments: Please avoid the duplicate use of Based on for two successive sentences.Also, How does surface morphology of the shale provide an indication of nanopore shape?]

Response 11: [The mineral analysis indicates that the Chang 7 shale is rich in organic kerogen and K-illite. The surface morphology of the Chang 7 shale suggests that the shape morphology of the nanopores can be simplified as slit-shaped in the molecular models.The organic pore is usually represented by the carbon-based pore] Thank you for pointing this out. We agree with this comment. Therefore, We have made the changes as requested.Mention exactly where in the revised manuscript this change can be found - page 3, paragraph 1, and line 1 to 4. 

Comments 12:[It is noted that two clay sheets arecontained in each simulation cell and each sheet has 24 unit cells, i.e., 6×4×1 supercell. In the supplementary Information, Figure S1 presents the schematic structure of the inorganic nanopore: K-illite. It is noted that we investigate the recovery mechanism of shale gas in organic and inorganic nanopores with two sizes Comment: Please delete It is noted from both sentences. It is a phrase that has no useful meaning.]

Response 12: Thank you for pointing this out. We agree with this comment. Therefore, We have removed these two sentences as per your request. 

Comments 13:[The TraPPE force filed Comment: This should be force field]

Response 13: [The force filed is used for ] Thank you for pointing this out. We agree with this comment. Therefore, We have made the changes as requested.Mention exactly where in the revised manuscript this change can be found - page 3, paragraph 2, and line 1 . 

Comments 14:[It is noted that the Ewald summation method is used Comment: Please omit It is noted]

Response 14: [The Ewald summation method is ] Thank you for pointing this out. We agree with this comment. Therefore, We have made the changes as requested.Mention exactly where in the revised manuscript this change can be found - page 3, paragraph 5, and line 1 . 

Comments 15:[In this study, the adsorption and desorption behavior of alkane/CO2 mixtures is investgated by the GCMC simulation method Comment: The research has been completed, so this statement should be past tense,. i.e., was investigated This comment applies to other present tense statements about the research]

Response 15: [the adsorption and desorption behavior of alkane/CO2 mixtures was investigated by the GCMC] Thank you for pointing this out. We agree with this comment. Therefore, We have made the changes as requested.Mention exactly where in the revised manuscript this change can be found - page 3, paragraph 6, and line 1 to 2 . 

Comments 16:[In the micropres and mesopores Comment: micropores and mesopores]

Response 16: [Micropres and mesopores, ] Thank you for pointing this out. We agree with this comment. Therefore, We have made the changes as requested.Mention exactly where in the revised manuscript this change can be found - page 4, paragraph 4, and line 4 to 5 . 

Comments 17:[In the micropres and mesopores, density of hydrocarbons in the pore surface

Comment: The description “in the pore surface” is curious, because be definition surface is a 2-

dimensional feature, so there can be no “in the surface”. Perhaps you mean “in the near surface”.]

Response 17: [Micropres and mesopores, density of hydrocarbons in the near surface is significantly higher than that in the pore center, indicating that hydrocarbons form adsorption layers on the pore surface,] Thank you for pointing this out. We agree with this comment. Therefore, We have made the changes as requested.Mention exactly where in the revised manuscript this change can be found - page 4, paragraph 4, and line 4 to 5 . 

Comments 18:[The density at the pore center of mesopores approaches that in bulk Comment: A clearer statement would be The density at the pore center of mesopores approaches the bulk density]

Response 18: [The density at the pore center of mesopores approaches the bulk density, ] Thank you for pointing this out. We agree with this comment. Therefore, We have made the changes as requested.Mention exactly where in the revised manuscript this change can be found - page 4, paragraph 4, and line 10 to 11 . 

Comments 19:[More interestingly Comment: I suggest that this is an inappropriate comment, as it implies that other parts of the presentation are lessinteresting.]

Response 19: [In addition, as for CH4-C4H10 mixtures, density of CH4 drops in the first adsorption layer as pressure decreases, ] Thank you for pointing this out. We agree with this comment. Therefore, We have made the changes as requested.Mention exactly where in the revised manuscript this change can be found - page 4, paragraph 4, and line 10 to 11 . 

Comments 20:[are tending to be trapped Comment: I think this would be better phrased as “tend to be trapped”]

Response 20: [ the heavier components tend to be trapped in these organic nanopores, ] Thank you for pointing this out. We agree with this comment. Therefore, We have made the changes as requested.Mention exactly where in the revised manuscript this change can be found - page 5, paragraph 1, and line 3 to 4 . 

Comments 21:[are tending to be trapped Comment: I think this would be better phrased as “tend to be trapped”]

Response 21: [ the heavier components tend to be trapped in these organic nanopores, ] Thank you for pointing this out. We agree with this comment. Therefore, [We have made the changes as requested.Mention exactly where in the revised manuscript this change can be found - page 5, paragraph 1, and line 3 to 4 . 

Comments 22:[In this process, CO2 is injected into the micro- and mesopores; the system pressure is increased from P1 to P2 during the huffing process and then it decreases to P3 in the puffing process.Comment: This would a useful description for the Abstract. It could be simplified to: In this process, CO2 is injected into the micro- and mesopores; the system pressure is increased during the huffing process decreased in the puffing process.]

Response 22: [The huff and puff process involves injecting CO2 into the micro- and mesopores, where the system pressure is increased during the huffing process and decreased during the puffing process. ] Thank you for pointing this out. We agree with this comment. Therefore, We have adopted your suggestion and added your expression to the summary. Thank you for your input.Mention exactly where in the revised manuscript this change can be found - page 1, paragraph 1, and line 5 to 7 . 

Comments 23:[Unlike that in organic pores, density in the adsorption layer of CO2 is much higher in the inorganic pore Comment : The term that is vague, it presumably refers to density of CO2. The wording can be improved.]

Response 23: [Unlike in organic pores, the density of CO2 in the adsorption layer within inorganic pores is significantly higher. ] Thank you for pointing this out. We agree with this comment. Therefore, We have made the changes as requested.Mention exactly where in the revised manuscript this change can be found - page 10, paragraph 1, and line 2 to 4 . 

Comments 24:[In this work, the GCMC simulation methods is applied Comment: Two small issues. The work has been completed, so the statement needs to be in past tense, e.g., was applied. Also, there is a singular/plural mismatch with methods is, which should be methods are.]

Response 24: [In this work, the GCMC simulation methods were applied to study ] Thank you for pointing this out. We agree with this comment. Therefore, We have made the changes as requested.Mention exactly where in the revised manuscript this change can be found - page 16, paragraph 1, and line 1 . 

Comments 25:[In addition to CO2 sequestration, CO2 mainly sequestrates in the adsorbed state in organic pores, while mainly in the free-gas state in the inorganic pores.Comment: The statement lacks clarity. What does the in addition to carbon sequestration refer to?]

Response 25: [Beyond traditional carbon sequestration methods, CO2 mainly sequestrates in the adsorbed state in organic pores, while mainly in the free-gas state in the inorganic pores.] Thank you for pointing this out. We agree with this comment. Therefore, We have made the changes as requested.This revision explicitly links the mechanisms of CO2 storage to the different pore types while clarifying that the discussion is about different sequestration contexts.Mention exactly where in the revised manuscript this change can be found - page 16, paragraph 1, and line 1 .

Comments 26:[In addition to CO2 sequestration, CO2 mainly sequestrates in the adsorbed state in organic pores, while mainly in the free-gas state in the inorganic pores.Comment: The statement lacks clarity. What does the in addition to carbon sequestration refer to?]

Response 26: [Beyond traditional carbon sequestration methods, CO2 mainly sequestrates in the adsorbed state in organic pores, while mainly in the free-gas state in the inorganic pores.] Thank you for pointing this out. We agree with this comment. Therefore, We have made the changes as requested.This revision explicitly links the mechanisms of CO2 storage to the different pore types while clarifying that the discussion is about different sequestration contexts.Mention exactly where in the revised manuscript this change can be found - page 16, paragraph 1, and line 1 .

Comments 27:[References: As is known, CO2 sequestration in the adsorbed state is safer than that in the free state Comment: This “free state” reference appears to answer to above question. However, the “It is known”declaration very much needs a reference.]

Response 27: We have added the cited references in the appropriate locations.Thank you for pointing this out. We agree with this comment. Mention exactly where in the revised manuscript this change can be found -  page 16, paragraph 3, and line 3 to 4 .

Comments 28:[References: Please take a careful look at the Instructions for Authors regarding format for reference entries, e.g., use of italic fonts, punctuation, capitalization, etc. There are variety of format inconsistencies.]

Response 28: We have revised the references as per the requirements.Thank you for pointing this out. We agree with this comment. Mention exactly where in the revised manuscript this change can be found - page 16 to 19.

Round 2

Reviewer 1 Report

Comments and Suggestions for Authors

All comments have been tackled.

Comments on the Quality of English Language

English could be rechecked.